# Digital Tools for Translucence Evaluation of Prosthodontic Materials: Application of Kubelka-Munk Theory

**DOI:** 10.3390/ijerph19084697

**Published:** 2022-04-13

**Authors:** Milagros Adobes-Martín, Natividad Alcón, María Victoria López-Mollá, Javier Gámez-Payá, Estibaliz López-Fernández

**Affiliations:** 1Faculty of Health Sciences, European University of Valencia, 46010 Valencia, Spain; milagros.adobes@universidadeuropea.es (M.A.-M.); mvlopezmolla@gmail.com (M.V.L.-M.); javier.gamez@universidadeuropea.es (J.G.-P.); estibaliz.lopez@universidadeuropea.es (E.L.-F.); 2Fisabio Foundation, 46020 Valencia, Spain

**Keywords:** color, digital image, material, optical properties, translucency, scattering, absorption, Kubelka-Munk, zirconia, visual matching

## Abstract

Translucency is one of the most important parameters to be considered by digital systems when predicting the matching appearance and hence the quality of prosthodontic restoration work. Our objective has been to improve the effectiveness of the algorithmic decision systems employed by these devices by (a) determining whether Kubelka-Munk theory can be used as an algorithm for predicting restoration suitability, and (b) evaluating the correlation between the visual evaluation of prosthodontic materials and the predicted translucency based on the use of the ΔE*, OP, CR, and K/S algorithms. In this regard, three zirconia systems and one lithium disilicate have been spectrophotometrically and visually characterized. Based on the results of this study, it has been proven that zirconia systems and lithium disilicate systems exhibit different optical behaviors. The psychophysical experience suggests that none of the existing mathematical methods can adequately estimate translucency, spectrophotometric, and colorimetric techniques, and that which is perceived by an experienced observer. However, translucency evaluation through the K/S algorithmic decision system should not be disregarded. New methods to measure translucency should be developed to improve digital systems for prosthodontic applications.

## 1. Introduction

There is no doubt that practically all the activities that we carry out on a regular basis, in one way or another, are linked to the use of different digital tools. In this context, perhaps prosthodontics is one of the branches of dentistry that has benefited the most from its use. In fact, the advantages of using CAD/CAM techniques in the development of personalized dental pieces are well known.

However, not only these digital tools are applicable in the prosthodontic field. Nowadays there are multiple devices that incorporate digital imaging systems with the aim to help with the restoration processes. These devices allow the dentist to have at the same time and in a unique screen, an image of the patient’s teeth and an image of the restorative material, helping in the comparison process and in the evaluation of the results.

All prosthodontic restoration procedures are intended to restore the function of the patient’s teeth. The use of materials that offer good adhesion and compatibility with the patient’s dental structure, guarantee the recovery of their function, and maintain the initial aesthetics of their teeth is a growing concern worldwide [1,2]. Thus, in order to achieve an optimal aesthetic result, the optical properties of the material and the patient’s teeth should be similar [1,3,4].

Despite the fact that different magnitudes have been used to describe the appearance of these materials such as color, degree of whiteness, and translucency [3,5], all of which are derived from their optical properties, translucency is the one with the greatest impact in dentistry. 

Without losing sight of the restoration objective, these digital tools may predict the perception match in terms of color and translucency using different algorithmic systems which are based on the spectrophotometric characterization of both materials: the patient’s tooth and the material used for the restoration. There have also been attempts to use intraoral digital scanners to evaluate these optical properties since some of them are able to use colored impression images obtained during the scanning. However, it has been reported the inaccuracy of the optical measurements conducted with this methodology.

Different studies have attempted to measure the translucency of these materials using the CIE color difference formula ΔE*, based on CIELab* coordinates [6,7,8,9,10,11] obtained by placing the sample on a white and on a black background [12]. Contrast ratio (CR), which is defined as the ratio between the tristimulus Y value of a sample placed on a black and on a white background [2,7], has also been used, as well as the opalescence parameter (OP), determined from a* and b* CIE coordinates of the samples placed on a black and on a white background [2,13,14].

Nonetheless, these systems characterize the material primarily via colorimetry and not through translucency, therefore not solving the problem of measuring the perceptual match between the patient’s teeth and the restorative material. On the other hand, the results are relative, since they relate only measurements made on black and white substrates. 

Specifically, translucency depends on light scattering and this property is directly affected by the size and numbers of pores inside the materials. 

Since translucency is a function of two key factors, the absorption and diffusion of light in a material (absorption and scattering) [15], some authors have considered that this could be evaluated using the absorption and diffusion coefficients (K and S), established in the Kubelka-Munk theory [16,17]. The theory is based on a simplified mathematical model that states that the final reflectance of a homogeneous material, of very fine thickness, can be considered as the result of two radiation fluxes, one incident and the other emergent.

The theory considers two light vectors (i and j) traveling on opposite directions, where K and S are absorption and scattering coefficients of the translucent material, respectively. An elementary layer of thickness (dx) is used to describe the changes in light fluxes in both directions. The decreases of the flux in the downward direction due to absorption and scattering of the downward flux are Ki dx and Si dx, respectively, and the increase in the flux in the downward direction due to scattering from the upward flux is Sj dx. Similarly, the decreases of the flux in the upward direction due to absorption and scattering of the upward flux are Kj dx and Sj dx, respectively, and the increase in flux in the downward direction due to scattering from the downward flux is Si dx [18].

The results of the equation to measure this effect allow us to calculate both coefficients from the reflectance and/or transmittance measurement and to establish a direct correlation with the material translucency.

Keeping this in mind, certain assumptions have been adopted: (1) the material is of a constant finite thickness, (2) illumination is diffuse and homogenous, (3) optical effects at its edges may be neglected, (4) pigment particles are uniformly distributed within the layer and are smaller than the elementary thickness, and (5) no reflection occurs at the surface of the translucent material [18,19].

The advantage of this theory is the possibility to calculate both coefficients from the reflectance and/or transmittance measurement [15,18,20], and its application in translucency evaluation allows the use of values directly related to the structure and composition of the material itself.

The theory has already been applied to the measurement of the translucency of certain materials for dental applications [21]. To date, studies have mainly focused on the optical behavior of resins used in dental fillings and colorants for dental prosthetics [22] and have not addressed the relationship between translucency and the material’s physical characteristics.

On the other hand, it is well known that only the visual evaluation of the translucency enables a final acceptance of the matching translucency for the prosthodontics pieces applied. Therefore, since instrumental measurements can detect small differences in terms of translucency, this procedure will only be adequate if it has a good correlation with the visual evaluation of the characterized materials. Despite the fact that the human eye is capable of perceiving small differences, and even if these are considered clinically relevant, the evaluation is subjective and therefore cannot be standardized [23]. There is evidence to suggest that perception varies between experienced professionals and even for the same person at different times [24].

In view of this situation, we have established two main objectives for this project. First, we applied an adaptation of the Kubelka-Munk theory to evaluate the scattering and absorption coefficients for translucency characterization of different prosthodontic materials and, consequently, determine if this theory could be applied by digital systems to predict restoration suitability. Our hypothesis was that absorption and diffusion coefficients (K and S), can be optimal algorithms for discriminating prosthodontic materials. The second objective has been to evaluate the correlation between the visual evaluation of the materials and the resulting translucency results using ΔE*, OP, CR, and K/S as decision algorithms and, by doing so, determine the current digital systems’ suitability for dental restoration work. 

## 2. Materials and Methods

### 2.1. Samples

In this study, three zirconia systems and one lithium disilicate material have been characterized. 

The first one, a tetragonal zirconia: IPS e.max Zir CAD LT, processed by CAD/CAM technique and with a theoretical 1200 MPa flexural strength (TZ); the second, a cubic/tetragonal hybrid zirconia: IPS e.max ZirCAD MT Multi, processed by CAD/CAM technique and with a theoretical flexural strength of 850 MPa (HZ); the third, a tetragonal zirconia VITA YZ HT Color with a theoretical flexural strength of 1230 MPa (THZ); and finally, a lithium disilicate IPS e. Max Press. LT, processed by injection technique with a theoretical flexural strength of 450 MPa (LD). Table 1 summarizes the materials characterized.

Samples material have been acquired in A1 color guide VITA, 10 not glazed squared 1.5 × 1.5 × 0.1 cm samples of each material for instrumental characterization (this size guarantees that all measurement area of the equipment is covered by the material to be characterized) and 10 not glazed incisive monolithic crowns obtained by CAD/CAM technique for visual evaluation. The thickness of the incisors crowns has been selected according to the manufacturer indications (1 mm for zirconia crowns and 1.5 mm for disilicate).

Nowadays, the thickness of zirconia ceramics in monolithic restorations can be less than 1 mm, guaranteeing their mechanical resistance. In this experiment, for both hybrid and tetragonal zirconia, 1 mm of thickness have been chosen since lower values provide a greater greyish effect to the restoration [25]. However, LD samples are of 1,5 mm because a higher thickness is needed due to its lower bending resistance.

This thickness decision was based on the commercial houses recommendations and on standardization purposes since previous research have demonstrated that are appropriate to guarantee at the same time aesthetics and resistance for both anterior and posterior sectors [13,25,26,27,28].

The selected samples allow us to evaluate the intrinsic translucency of the materials studied, avoiding the inference of other materials and/or finishes that could interfere in their measurements and perception. 

A number of 10 samples was chosen based on the following criteria: a) the materials analysed are very homogeneous, so the variation in the samples studied is very small. This was corroborated by the low data dispersion obtained from the standard deviation and the interquartile range, and b) this study followed a similar methodology to other studies of the same field, in which a sample size of less than 10 was included [15,29,30].

Moreover, through G*Power we calculated the statistical power a posteriori and we found a power of 0.806, which is considered acceptable for this study.

### 2.2. Reflectance/Transmittance Measurements

In order to characterize the dental ceramics with the aim to evaluate their translucency, it is possible to use different techniques: direct transmission (measuring the light that reaches a detector); total transmission (measuring both the light that reaches the detector and the one that passes the ceramic and is scattered); and indirect measurements via spectral reflectance [16]. In this study, we have used the spectral reflectance method to evaluate samples’ translucency.

The samples have been characterized using a double beam UV-V spectrophotometer (Lamba 35 UV/V, Perkin Elmer, Waltham, MA, USA), equipped with an integrating sphere, with circular measurement area of 1 cm^2^.

The measurement geometry was 0/d (specular component included) and the scans were carried out between 380 nm and 780 nm, with a bandwidth of 2 nm.

Before conducting the measurements, the equipment has been set-up following the equipment’s specifications, using the white adjustment plate provided by the manufacturer for its calibration.

Applying this technique, the reflectance (Rλ) of the 10 samples of the four materials (TZ, HZ, THZ, and LD) has been obtained under the following conditions: (a) Rλ_B_ has been measured by placing each sample on a black background (b) Rλ_W_ has been measured by placing each sample on a white background.

Both the black and white substrates have also been characterized through their Rλ.

The zirconia and lithium disilicate samples have been measured three times, each one of them, in different sessions.

### 2.3. CIELab* Coordinates

Once the Rλ of each sample has been obtained under the conditions described above, CIELab* coordinates and tristimulus Y values have been calculated for Illuminant D65 and CIE 10° standard observer.

### 2.4. CIELab* Color Difference (ΔE*)

ΔE* was obtained using the following formula:(1)$$\Delta E=\sqrt{{\left(L_{b}^{*}-L_{w}^{*}\right)}^{2}+\left(a_{b}^{*}-a_{w}^{*}\right)+{\left(b_{b}^{*}-b_{w}^{*}\right)}^{2}}$$
where L*_b_, a*_b_, and b*_b_ are the chromatic coordinates CIELab* over black background and L*_w_, a*_w_, and b*_w_ are the chromatic coordinates CIELab* over white background [2,6].

### 2.5. Contrast Ratio (CR)

CR was obtained using the following formula:(2)$$CR=\frac{Y_{b}}{Y_{w}}$$
where Y_b_ is the tristimulus value Y obtained when the samples are placed over black background and Y_w_ is the tristimulus value Y obtained when the samples are placed over white background [2,6].

### 2.6. Opalescence Parameter (OP)

OP was obtained using the following formula:(3)$$\mathrm{OP}=\sqrt{\left(a_{b}^{*}-a_{w}^{*}\right)+{\left(b_{b}^{*}-b_{w}^{*}\right)}^{2}}$$
where a*_b_ and b*_b_ are the chromatic coordinates CIE a* and b* over black background and a*_w_ and b*_w_ are the chromatic coordinates CIE a* and b* over white background [2,8].

### 2.7. Kubelka-Munk Coefficients

Once the Rλ of each sample was obtained under the conditions described before and following the methodology presented by M. Perez and E. Hita [15,21,31], the Kubelka-Munk scattering coefficient (S) and absorption coefficient (K) were calculated from the spectral reflectance data using the Kubelka-Munk equations.

The scattering coefficient (S) has been calculated, for a unit of thickness of a specific material, following this equation:(4)$$S\left({\mathrm{mm}}^{-1}\right)=\frac{1}{bX}\;arctgh\left[\frac{1-aR_{0}}{bR_{0}}\right]$$
where X is the thickness of the specimen, and arctgh is an inverse hyperbolic cotangent [15].

The absorption coefficient (K) is calculated as:(5)$$K\left({\mathrm{mm}}^{-1}\right)=S\left(a-1\right)$$

Secondary optical constants (a and b) were calculated from the experimentally obtained spectral reflectance values for the black and white background, using the following equations:(6)$$a=\frac{1}{2}R+\left[\frac{R_{0}-R+R_{g}}{R_{0}\;R_{g}}\right]$$
(7)b=(a2−1)12
where R_g_ is the reflectance of the white background, R_0_ is the reflectance of the specimen over the black background, and R is the reflectance of the specimen over the white background [15].

Finally, the ratio K/S, proportional to the ratio of the scattering and absorption coefficients was also calculated.

### 2.8. Visual Evaluation

Data was collected from 15 dental professionals, all of them dentists, previously informed and with a consent signature, 4 males (M) and 12 females (F), age ranging from 25 to 61 years old, with normal color vision (evaluated using Ishihara’s Test for Color-Blindness), and with some knowledge and previous clinical experience with shade matching. To determine the sample size for the visual evaluation, we use a review presented in 2015 (Schmidt et al. 2015) [32]. In this paper, the methodological basis for conducting an analytic hierarchical process was presented. Regarding the sample size, this article concluded that the number of participants can vary from 4 to more than 100 people, depending on the features of the experimental sample. The more homogeneity of the sample, less participants are needed [32].

Since our visual inspection was conducted by a very homogeneous group of dental professionals which guarantees a low dispersion of results, we decided to include 15 participants with the same profile in the research to conduct the visual inspection.

The visual experiments were performed in two phases and each of them were conducted under two different light sources. One corresponding to an illuminant with a color temperature of 2750 °K and the other corresponding to an illuminant with a color temperature of 4000 °K [6,33]. The evaluations were done within a distance of 35 cm from the samples that were placed at neutral gray surround (avoiding specular reflection from the glossy surface).

In phase I, the evaluators were asked to arrange each of the four materials from the highest to the lowest value of translucency according to their visual perception.

In phase II, the samples were shown in pairs, requesting the evaluators sort out which one had more translucency and assess that difference on a scale from 0 to 9. This was conducted following the SAATY Analytic Hierarchy Process [34].

This evaluation involves the participation of several dental professionals, needing therefore the approval of the Ethics Committee of the European University. It was conducted according to the Consolidated Standards of Reporting Trials (CONSORT) Statement and Helsinki Declaration of 1975, revised in 2000, and the present Spanish law (Ley 14/2007, de 3 de julio, de Investigación Biomédica).

### 2.9. Statistical Analysis

The distributions of data did not meet the parametric assumptions. Thus, Kruskall–Wallis test was used to determine the difference K/S values among materials. To conduct the post-hoc analysis, we used Mann–Whitney U test. A 0.05 level of significance was adopted. The aforementioned analysis was conducted using SPSS (IBM Corp. (Armonk, NY, USA) Released 2016. IBM SPSS Statistics for Windows, Version 24.0. IBM Corp: Armonk, NY, USA). Lastly, we determined the size effect using the approach proposed by Rosenthal (1991) (low effect: 0.1; medium effect: 0.3; large effect: 0.5) [35].
(8)$$r=\frac{z}{\sqrt{N}}$$

## 3. Results

### 3.1. Reflectance/Transmitance

In Figure 1, the reflectance/transmittance spectra of the four materials are presented as the mean of all measurements in terms of Rλ_B_ and Rλ_W_.

The spectral behavior of the three zirconia is very similar, showing a lower peak at 650 nm and 520 nm similar for Rλ_B_ and for Rλ_W_. This fact is not observed with the LD material, where the distribution is more uniform in all the spectrum.

In the range of 380–440 nm, TZ, HZ, and THZ show lower values than LD.

If we compare the three zircon systems from 440 to 780, THZ has higher values of reflectance, followed by TZ and HZ. This fact is similar for Rλ_B_ and for Rλ_W_.

### 3.2. K/S Ratio

The values of K/S in function of the wavelength are shown in Figure 2.

According to the direct relationship between Rλ and K and S values, the results obtained follow analogous trends. Based on the K/S values, the curves for the three zirconia materials follow a similar pattern, which is understandable when dealing with similar materials, and they differ from lithium disilicate.

In order to have a better interpretation of the data, the curves have been divided into two segments: firstly, the section between 380 nm and 470 nm, and secondly, the section between 480 nm and 780 nm (Figure 3).

A comparative study between materials can give us more detailed information about the significance of the values obtained. The statistical comparative study between materials as well as the size effect among materials shows statistical differences in every comparison among materials except when we compared K/S ratio in TZ and THZ at 380 nm where we found a non-significant trend (*p* = 0.054). Despite that, it is important to highlight that we found a large and medium size effect in all analysis conducted among materials.

Regarding the first section, there are significant differences between the K/S values for both types of materials, confirming the validity of these differences through statistical data analysis (*p* < 0.05). Differences between both tetragonal zircons are significant from 400 nm (*p* < 0.05), but not for lower wavelength.

In this section, HZ presents higher values in K/S ratio than the tetragonal zircons, enough to differentiate both materials, and LD has the lowest values (*p* < 0.05).

For the second section between 480 nm and 780 nm, a similar trend is observed. The behavior of the three zirconia samples follows a similar pattern to that observed for the segment between 380 nm and 470 nm; however, here the differences are less pronounced (*p* < 0.05). There is a distinct difference between the behavior of LD and that of the first analyzed section.

Obviously, these results are directly related to the values obtained for the absorption coefficient (K) and scattering coefficient (S).

### 3.3. CIELab* Color Difference ΔE*), Contrast Ratio (CR) and Opalescence Parameter (OP)

Table 2 summarizes the results obtained for the CIELab * Color difference ΔE*, CR, and OP for each material. The results are the average of all the measurements performed.

Table 3 shows the classification according with the optical parameters used in terms of translucency. The ranking obtained for CR and ΔE* is identical.

In relation with opalescence evaluation, the ranking in terms of OP is notably different. OP parameter does not consider the luminosity of the samples. OP calculation only considers the values of chromatic coordinates a* and b* that give information about the more or less red, green, yellow, or blue in the color of the samples, but it does not consider the L* value. This is not the same for the calculation of CR and ΔE* values.

On the other hand, it was also statistically evaluated if CR, ΔE*, and OP were good parameters to establish the difference in translucency when compared two by two.

Using a statistical comparative study, we have observed statistically significant differences in every comparison except between LD and TZ, and HZ and THZ when compared in terms of ΔE*; LD and HZ when compared in terms of CR; and LD and THZ when compared in terms of OP (*p* > 0.50). Despite that, it is important to point out that large and medium-sized effects were found in all analyses between materials.

### 3.4. Visual Evaluation

Figure 4 and Figure 5 represent the results of the visual evaluation conducted by the 15 volunteers.

Figure 4 shows an order of the highest to the lowest values of translucency perceived by the volunteers when the four samples were compared at once under both types of light sources. 

Interestingly, no observer perceived the TZ material as most translucent, regardless of the lighting conditions. However, the results are not as clear for LD because some observers (40% under light source of color temperature 2750 °K and 33% under light source of color temperature 4000 °K) classified this material as being the most translucent, while others classified it with the lowest translucency (53% under light source of color temperature 2750 °K and 40% under light source of color temperature 4000 °K).

Finally, Figure 5 shows the results obtained when Saaty’s analytical hierarchies method was performed in the weighting of variables. The experiment consisted in the evaluation of all materials arranged in pairs under both light sources explained above.

The results obtained indicate that translucency evaluation is similar in both scenarios and those results are consistent according to the Saaty’s method (Table 4). The hierarchy obtained is the following:

As a summary, visual evaluation indicates the most translucent material perceived by the volunteers is HZ, and the lowest is TZ.

## 4. Discussion

As we have mentioned before, a correct dental restauration implies a visible match of the optical properties of prosthodontic materials and natural teeth. The optical properties depend on the absorption and scattering process of the light, which result from the interaction of the light with the materials.

Keeping this in mind, our goal in this study has been to demonstrate that the two coefficients (K and S), deriving from the Kubelka-Munk theory, can be used to evaluate the translucency of different prosthodontic materials and to evaluate the correlation between the visual evaluation of these materials and measurement of translucency in terms of ΔE*, OP, CR, and K/S. Commercial materials used in the dental practice, such as different zircons (tetragonal, cubic/tetragonal, and last generation tetragonal) and a Lithium Disilicate ceramic were selected. 

In regards with optical properties, the color and appearance of dental restorative materials depend upon the environment in which they are applied (characteristics of the substrate to be restored, type of cement, etc.) and on their intrinsic properties (grain size, pores, etc.) [22,36]. A direct relationship between material-light interaction and the passage of light through a medium with a different refractive index is well known; in this case grain/grain, grain/pore, etc. It should be noted that diffusion, transmission, reflection, absorption, and refraction, and thus its color, translucency, and opacity, are dependent on the microstructure and chemical composition of the material [30]. In this context, it is also important to consider the role that glazing procedures of dental pieces could play in obtaining certain optical properties. It is evident that the interposition of a new medium between the light and the material with a different refractive index and structure, will modify light interaction and therefore influence the physical appearance of the dental piece.

Concerning materials, lithium disilicate has been one of the most widely used ceramic materials in dental restoration during the last decades. It is classified as glass ceramic and its formulation can be modified by varying the proportions of its constituents, such as ZnO, ZrO_2_, CaO, and P_2_O_5_, in order to improve its properties and to modify the formation and crystallization phases. For example, small amounts of P_2_O_5_ added in its composition will give a fine-grained microstructure with higher mechanical strength [37].

Nowadays, research is focused on the microstructure level in order to improve mechanical and optical properties of new biomaterials [22]. In recent years, cubic/tetragonal zircons of the so-called last generation have been introduced, with good mechanical properties and improved aesthetic features in comparison to their predecessors (other cubic/tetragonal zircons and tetragonal zircons) [38,39,40,41,42]. Dental zirconia is a polymorphic, polycrystalline ceramic. It shows three phases, or crystal structures, with specific geometry and dimensional parameters: monoclinic, tetragonal, and cubic. The mechanical and optical properties change in function to the crystalline structure. Different oxides (Y_2_O_3_, CaO, MgO, etc.), can be added to stabilize the zirconia and to allow behavioral modifications of the three phases. The composition and the manufacturing process will directly affect its mechanical and optical properties resulting in ceramics with improved characteristics, such as high flexural strength and fracture toughness, high hardness, excellent chemical resistance, and good conductivity ions. The sintering final temperature, the atmospheric conditions during the sintering process, and the heating methods are directly related with density, porosity, and grain size [30,43]. Different studies have shown a correlation between small grain size with enhanced translucency, better mechanical properties, and a delay degradation when tetragonal and monoclinic structures are compared [30]. All this together may explain the differences observed in the optical properties between lithium disilicate and zirconia [3,4].

The first parameter studied has been the spectral reflectance of the materials in terms of R_w_ and R_b_. The behavior was very similar between the three zircons, and different from DL at short wavelength.

The similar behavior of the three zircons may be justified by their structural composition. The decreased values of R_w_ and R_b_ observed at low wavelength could be a correlation between the pore size of the zircons and the wavelength of the incident light. Different authors have demonstrated that light scattering is dependent on the grain size and wavelength of the incident light. If the grain size is similar to the wavelength of incident light, the light scattering increases with grain size. If grain size is much larger than the wavelength of incident light, the amount of light scattering becomes inversely proportional to the grain size, and independent of the wavelength of incident light and the transmittance increase [30,44]. LD has a more uniform behavior probably due to a more consistent pore size along the material examined. Nevertheless, this appreciation must be confirmed by microscopically measurement of the pore size that have not been considered in this study.

Knowing that translucency is a complex phenomenon, conditioned by the processes of reflection, transmission, diffusion, and absorption of materials, both internally and externally, we have also studied the relationship between K/S and translucency in order to know if it could give more information about the level of dental material translucency. It may be assumed that low K/S values give higher levels of translucency since increased levels of S is correlated with greater light diffusion and therefore a greater capacity of the material to show better translucency [15]. This assumption should be endorsed by the perception analysis of these materials that will be addressed later on.

As we have mentioned, in order to better analyze the data, the curves have been divided into two segments, the first one compromises the wavelengths between 380 nm and 470 nm, and the second one between 480 nm and 780 nm.

As shown by data obtained from the first portion of the spectrum (380 to 470 nm), K/S ratio differentiates both types of materials, lithium disilicate and zirconia, well. It shows a predominance of scattering (S) that is even greater for LD material. This result is in concordance with some reported experiences that indicated the higher influence of S in the translucency of dental materials [15]. The tendency of S values obtained through the spectra, corroborates the result obtained by Fernandez-Oliveras et al. [16]. Considering that a lower value of K/S means a higher level of translucency, our study in this first portion of the spectra shows that LD is the most translucent material followed by THZ, TZ, and HZ.

In terms of translucency, and according to the literature, HZ is at least theoretically more translucent than tetragonal zircons, and TZ and THZ should have similar values and, if not, TZH should be higher than TZ [45]. Our results do not completely agree with this statement, even though they also demonstrate that the new generation of hybrid zirconia are more translucent than the tetragonal ones. In fact, the U Mann–Whitney parameter evidenced differences between all the materials when pairwise comparison is made.

In the second section of the spectrum (480–780 nm), the differentiation between both materials by K/S is statistically confirmed, as well as among the three zirconia. Even though the results obtained for the zircons are similar with the ones obtained in the first section of the spectrum, in this case LD shows to be the less translucent material. This result is in concordance with Baldissara et al.’s [46] research.

Nevertheless, it is necessary to conduct more studies to explain the different behavior of the materials related to their crystalline structure and the characteristics of the incident light [30,45], since pore/grain size and wavelength directly interfere with diffusion and scattering effects [44].

This ordering, the most translucent THZ followed by HZ and the last one LD, is also observed when CR and ΔE* parameters are analyzed. These results agree with Baldissara et al.’s [46] research, which indicated that the suppression of the tetragonal phase improves translucency and that even hybrid zircons outperform LD in this property.

According to the different parameters, the translucency position of TZ varies in relation to the other materials, but the results are not statistically significant for all of them, so no conclusion can be drawn.

The goodness of these results obtained so far in terms of translucency assessment could only be assumed if experienced observers order the prosthetic materials in the same way as done with the experimental measurements.

Considering all the environmental conditions that could affect a visual evaluation [47], statistical analysis of the results obtained from visual perception confirmed the ones obtained at the experimental phase. This is also supported by the coincidence of ordering between the evaluations conducted under the two light sources considered in the experience.

Interestingly, TZ seems to be the material perceived with less translucency as it was when CR and ΔE* were used for their characterization. However, HZ was determined as more translucent than THZ when the data obtained with CR, ΔE*, and K/S point out in an opposite direction.

These arrangement differences showed that there is not a perfect correlation between the instrumental evaluation and the perception of optical properties, and the difficulty in obviating the subjective behavior associated to the cognitive abilities of the evaluators.

As far as possible, the work has tried to standardize the systematics followed in the comparison process. For this reason, the surroundings have been controlled, the minimum requirements have been established regarding the perception of the color of the participants, and the evaluators have been a group of professionals with similar dental experience, as some authors indicate it is an important factor to consider, given the subjectivity of the evaluation [48].

In this case, the differences in color and translucency between materials exceed the thresholds that the literature reflects as perceptible minimums between samples of different translucency [23,48].

The experience conducted evidenced that the quantification of translucency through the parameters used (ΔE *, OP, CR, and K/S) does not allow a perfect correlation with the perception of it.

Consequently, in the future, new research must be conducted in order to develop new algorithms or introduce modifications to the existing ones, in order to achieve a perfect translucency measurement system.

It is difficult to predict what the solution to the problem may be. The new method must correlate the processes of absorption, reflection, and diffusion of light in the material with its chemical and morphological structure (presence of pores, grain size, crystalline structure, etc.), as well as with the neuronal processes derived from the activity of the photoreceptors present in the retina and involved in the process of visual perception.

Ideally, future research have to avoid the limitations of previous studies (materials, characteristics of the samples characterized, assay methodology, etc.) in order to develop a useful algorithm.

## 5. Conclusions

Within the limitations of this study, we have evidenced differences in optical behavior between zirconia systems and lithium disilicate. Parameter K/S, determined in a simplified mode, could be valid in order to establish a differentiation between both materials.

The psychophysical experience conducted would suggest that none of the existing mathematical methods allow a good translucency assessment between spectrophotometric and colorimetric techniques with that perceived by an experienced observer. Additionally, translucency evaluation through the K/S algorithmic decision system should not be rejected since its results showed deficiencies similar to those obtained using the hitherto classics CR, OP, and ΔE*. 

This supports the idea that new methods to measure translucency should be developed, just so digital systems for prosthodontic applications could be improved and be established as an effective aiding tool for the odontologist.

## Figures and Tables

**Figure 1 ijerph-19-04697-f001:**
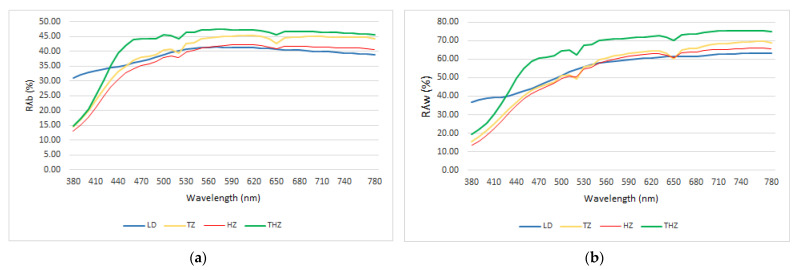
Reflectance spectra of the three samples, as mean of all measurements: (**a**) Reflectance spectra in terms of Rλ_B_; (**b**) Reflectance spectra in terms of Rλ_W_.

**Figure 2 ijerph-19-04697-f002:**
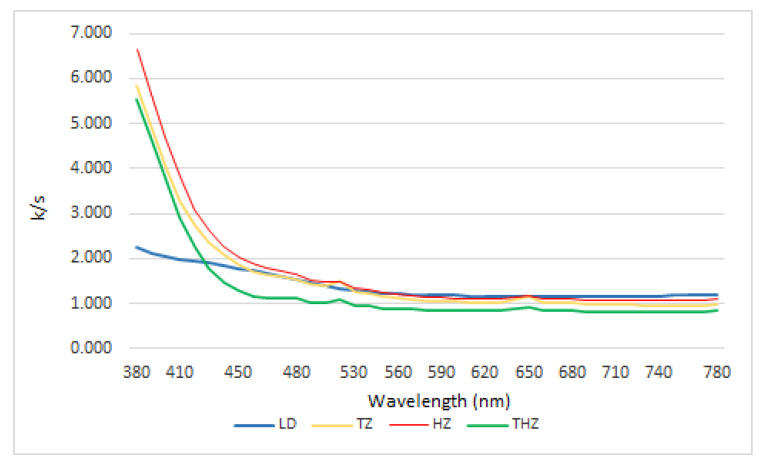
Spectral distribution of the K/S ratio for three zirconia and lithium disilicate. Wavelength range from 380 to 780 nm.

**Figure 3 ijerph-19-04697-f003:**
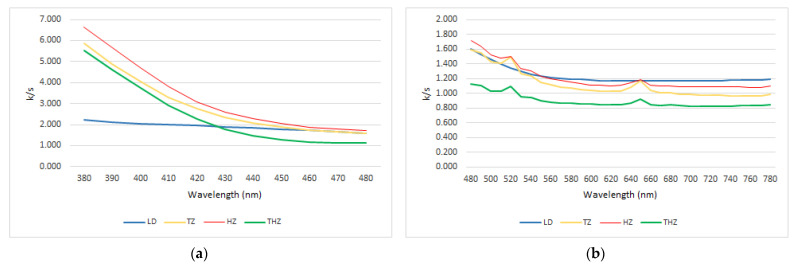
Spectral distribution of the K/S ratio for three zirconia and lithium disilicate. (**a**) Wavelength range from 380 to 470 nm. (**b**) Wavelength range from 480 to 780 nm.

**Figure 4 ijerph-19-04697-f004:**
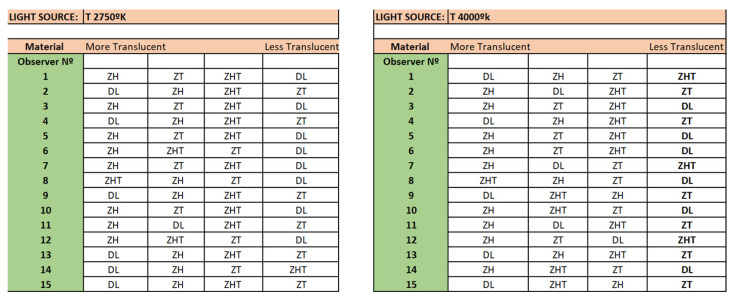
Visual evaluation of the four materials under both light sources.

**Figure 5 ijerph-19-04697-f005:**
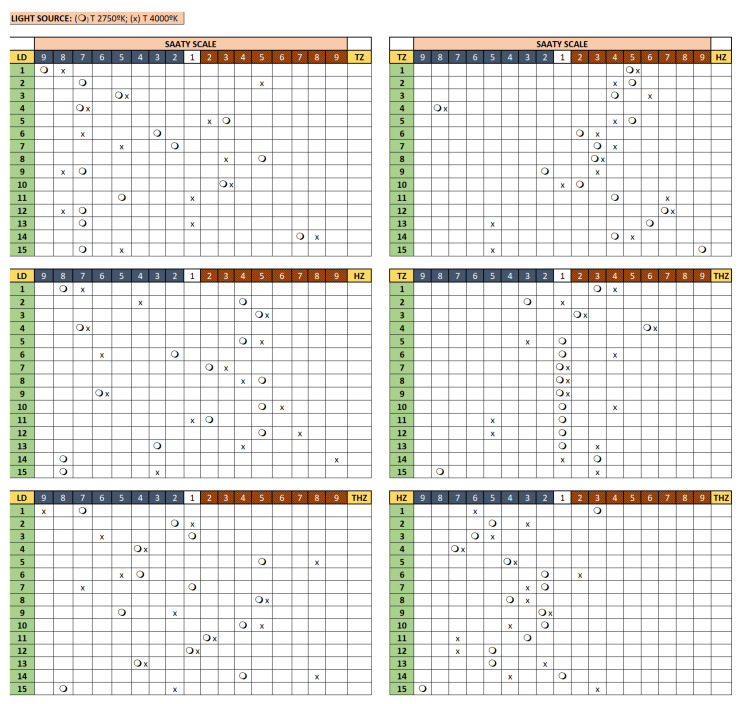
Results obtained after Saaty’s analysis when the materials where paired compared.

**Table 1 ijerph-19-04697-t001:** Characterized materials and their acronyms.

MATERIAL	ACRONYM
Tetragonal Zirconia: IPS e.max Zir CAD LT	TZ
Cubic/Tetragonal Hybrid Zirconia: IPS e.max ZirCAD MT Multi	HZ
Tetragonal Zirconia VITA YZ HT Color	THZ
Lithium Disilicate IPS e. Max Press. LT	LD

**Table 2 ijerph-19-04697-t002:** Average measurements of ΔE*, CR, and OP for each material studied.

	LD	TZ	HZ	THZ
CR	0.71798	0.74971	0.7161	0.6818
OP	7.6344	8.7323	9.4037	7.2601
ΔE*	12.5863	12.4159	13.7695	14.2007

**Table 3 ijerph-19-04697-t003:** Translucency rankings of all materials under study according to the results obtained in Table 2.

HIGHEST TRANSLUCENCY	CR	ΔE*
	THZ	THZ
	HZ	HZ
	LD	LD
LOWEST TRANSLUCENCY	TZ	TZ

**Table 4 ijerph-19-04697-t004:** Hierarchy after Saaty’s analysis.

T 2750 °K	Material	Weight	T 4000 °K	Material	Weight
	HZ	0.38		HZ	0.39
	LD	0.31		LD	0.27
	THZ	0.17		THZ	0.18
	TZ	0.14		TZ	0.16

## Data Availability

Not applicable.

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
