# Peer review of "Digital Tools for Translucence Evaluation of Prosthodontic Materials: Application of Kubelka-Munk Theory"

_ijerph, 2022, doi:10.3390/ijerph19084697_

Round 1
Reviewer 1 Report
Dear Authors, Thank you for your study. Overall, the manuscript is well prepared. There are some minor issues that should be addressed. Please insert a table that defines the abbreviations TZ, HZ, THZ, LD. Further, why an ethical approval is necessary. Please comment. If not appreciable please delete this paragraph from the M&M section. In addition, please provide the reader with more information on the Kubelka-Munk theory and why it is important for this study.
Reviewer 2 Report
Dear authors, thank you for submitting your manuscript entitled "Digital tools for translucence evaluation of prosthodontic materials. Application of Kubelka-Munk theory".
After a detailed evaluation, I have found many areas for major improvements:
In the materials and methods section, please do not start a sentence with a number (20). A native English speaker needs to review the entire manuscript.
You have selected thickness of 1.0 mm for zirconia and 1.5 mm for lithium disilicate, which are the previous recommendation for crowns, however why you did not select thinner samples in order to relate your study to veneer restorations ?
Also you selected 1.5 mm for lithium disilicate but that is the old recommendation from the company. Since 2018, the recommendation for bonded crowns is 1.0 mm. Why you did not include the 1.0 mm recommendation for lithium disilicate ?
Why did you select 1.0 mm thickness for zirconia, since some companies accept 0.8 mm and 0.7 mm thickness for crowns ?
Explain why you sample size is 1.5 x 1.5 x 0.1 cm.
How the glaze was applied ? technique for application and amount.
You only have 10 samples per group, is this an acceptable number ? please do a G* Power Data Analysis for that.
You used a spectrophotometer that is ideal for petro-chemical laboratory uses (according to the website company). Why you did not use a dental spectrophotometer ?
There are very advanced dental spectrophotometer in the market such as Spectroshade 2 that can provide all measurements including translucency. Please include the information of those available tools and why you did not use them.
For visual evaluation, you mentioned dental professional, please describe what type of dental professionals (dentists, technicians, hygienist, residents, or students). Why the number of 15 people?
Your figures will look better if you provide a specific color for each group (easier to visualize). You can also combine table 1 and 2 because they have basically the same information.
You have too many figures that are plain and unattractive for the lecturer, please improve their style.
Thanks.
Reviewer 3 Report
" Digital tools for translucence evaluation of prosthodontic materials. Application of Kubelka-Munk theory."
It is very interesting to evaluate whether the Kubelka-Munk theory can be used digitally and the correlation with the translucency results obtained using ΔE *, OP, CR, and K / S as deterministic algorithms. However, there are a few corrections that are essential to meet the standard for publication. Please refer to the following comments.
- The authors conclude that a new method for measuring translucency needs to be developed. Please add to the discussion about the prospect of a concrete new method.
- Please add your research limitations in the discussion section. Showing the limitations of this study will be useful information for many researchers who will be responsible for the next study.
- Many references have been adopted in your manuscript. Aside from comparisons with past reports, please update the citations as new as possible for the latest findings.
- Unfortunately, all figures do not have X-axis and y-axis headings and units. Please add them.
Round 2
Reviewer 2 Report
Dear Authors,
Thank you for your manuscript entitled "Digital tools for translucence evaluation of prosthodontic materials. Application of Kubelka-Munk theory".
I want to congratulate you, because I see an important improvement in your manuscript. However, there are minor points that need to be addressed.
Your samples were not glazed, and some technicians/clinicians prefer to glaze restorations, please include this information in the discussion and the rational or the variable that could cause glazing the samples.
It will be interesting for the reader to have spectrophotometer images of the samples because it will be easier and simpler for a clinician to follow the results. Please consider including them.
I know chairside dental scanners can evaluate tooth shades in a broad aspect, however I am not aware about they can measure the translucency and opalescence, please search for this information and mention it in the introduction.
Thank you.
